# Single, Subsequent, or Simultaneous Treatments to Mitigate Mycotoxins in Solid Foods and Feeds: A Critical Review

**DOI:** 10.3390/foods11203304

**Published:** 2022-10-21

**Authors:** Alaa Abou Dib, Jean Claude Assaf, André El Khoury, Sami El Khatib, Mohamed Koubaa, Nicolas Louka

**Affiliations:** 1Centre d’Analyses et de Recherche (CAR), Unité de Recherche Technologies et Valorisation Agro-Alimentaire (UR-TVA), Faculté des Sciences, Campus des Sciences et Technologies, Université Saint-Joseph de Beyrouth, Mar Roukos, Matn 1104-2020, Lebanon; 2Department of Food Sciences and Technology, Facuty of Arts and Sciences, Bekaa Campus, Lebanese International University, Khiyara, Bekaa 1108, Lebanon; 3TIMR (Integrated Transformations of Renewable Matter), Centre de Recherche Royallieu, Université de Technologie de Compiègne, ESCOM—CS 60319, CEDEX, 60203 Compiègne, France

**Keywords:** mycotoxins, mitigation treatments, decontamination rates, single treatments, combined treatments

## Abstract

Mycotoxins in solid foods and feeds jeopardize the public health of humans and animals and cause food security issues. The inefficacy of most preventive measures to control the production of fungi in foods and feeds during the pre-harvest and post-harvest stages incited interest in the mitigation of these mycotoxins that can be conducted by the application of various chemical, physical, and/or biological treatments. These treatments are implemented separately or through a combination of two or more treatments simultaneously or subsequently. The reduction rates of the methods differ greatly, as do their effect on the organoleptic attributes, nutritional quality, and the environment. This critical review aims at summarizing the latest studies related to the mitigation of mycotoxins in solid foods and feeds. It discusses and evaluates the single and combined mycotoxin reduction treatments, compares their efficiency, elaborates on their advantages and disadvantages, and sheds light on the treated foods or feeds, as well as on their environmental impact.

## 1. Introduction

In a world full of economic, health, and environmental crises, food security concerns have become one of the most important dilemmas of our era. Fungal infection and the resulting production of mycotoxins in crops are major problems caused by climate change as a result of global warming [1,2]. Cereals and grains are considered highly susceptible to such types of infection during the pre-harvest and post-harvest stages of their production; their availability has a vital role in preventing hunger and food insecurity [3]. Mycotoxins, the secondary metabolites of fungi, are considered a food safety challenge, threatening the lives of humans and animals due to their immune toxicity, carcinogenicity, hepatotoxicity, nephrotoxicity, mutagenicity, and teratogenicity [4,5]. The pathogenicity and the toxigenic potentials of many fungal species, such as *Aspergillus*, *Claviceps*, *Fusarium*, *Penicillium,* and *Alternaria,* have been reported in various crops [6]. Their occurrence and presence in a specific food product at a specific geographic region depend on extrinsic factors related to environmental conditions fluctuation, such as temperature and relative humidity, which explain the effect of global climate change on the formation of these mycotoxins in agricultural commodities [7,8,9]. The type and the number of mycotoxins in foods and feeds are directly related to many intrinsic factors, such as the moisture content, the pH, the composition of the food, and many other extrinsic factors, such as the relative humidity and the storage temperature [10]. The most commonly known mycotoxins to contaminate foods and feeds are aflatoxins AFs (AFB_1_, AFB_2_, AFG_1_, AFG_2,_ AFM_1_), ochratoxin (OTA), trichothecenes (deoxynivalenol: DON, nivalenol: NIV, T-2 toxin: T-2, HT-2 toxin: HT-2), zearalenone (ZEN), fumonisins B1 (FB1), enniatins (EN) [11,12,13], moniliformin (MON), beauvericin (BEA), and fusaproliferin (FUS) [14,15].

According to Food and Agriculture Organization (FAO) reports, 25% of the crops in the world are contaminated by mycotoxins [5,16,17]. Eskola et al., found that this percentage is underestimated and that 60 to 80% of crops are contaminated by mycotoxins [18]. In the United States, aflatoxin contamination causes great losses in the corn industry, reaching up to USD 1.68 billion. According to the Rapid Alert System for Food and Feed (RASFF), most rejection notifications at the EU border are due to mycotoxin contamination [19]. Regulations for mycotoxins are not available worldwide, especially in African countries. Mycotoxins in food and feed are extensively regulated in Europe. At the same time, aflatoxins in foods, particularly AFB_1_, are the most commonly regulated mycotoxins in many countries. Total aflatoxin limits in food were established in 2003 in 48 countries [20,21]. The maximum acceptable levels of the total AFs are 4μ/kg in the European Union and 20 μg/kg in the United States [16]. Globally, the maximum levels for AFs (B_1_, B_2_, G_1_, and G_2_), AFM_1_, and OTA in food are regulated by the codex standard CXS 193-1995 and established by the Codex Alimentarius Commission of the Food [18].

*“Prevention is better than cure”*, and this should be the first strategy for reducing mycotoxins in feeds and foods [22]. This preventive strategy aims at controlling the fungal growth and the production of these metabolites in foods and feeds in the pre-harvest and post-harvest stages by applying good agricultural practices and monitoring storage and processing conditions [23,24]. Practically, complete prevention of the formation of mycotoxins in crops is not feasible, which has triggered the need for alternative strategies aiming at decreasing or eliminating the amount of already produced mycotoxins in food and feed materials [25,26,27].

Chemical, physical, and biological technologies and treatments have been established and studied to mitigate mycotoxins in foods [28,29]. The success and efficiency of the method used to reduce mycotoxins depend on the food or feed characteristics [10]. Many studies have provided insights on a number of techniques designed for the detoxification of mycotoxins in liquid medium [30,31], such as in milk and dairy products using lactic acid bacteria biofilm [32], chitin, and shrimp shells [33] or by chemical treatment, such as ozonation [34] in fruit juices and wine [35,36] and in solid foods and feeds [37,38]. These technologies can be implemented separately, one by one, or combined in order to attain additive or synergistic effects in the reduction of mycotoxins in food or feed.

In this review, we focus on mitigating mycotoxins in solid foods and feeds only. We evaluate the effectiveness of chemical, physical, and biological treatments applied to solid foods and feeds to reduce mycotoxins when implemented separately and/or subsequently or simultaneously combined, as shown in Figure 1. In addition, we evaluate the effect of the different treatment modalities on the quality of the treated food materials providing their advantages and disadvantages.

## 2. Single Detoxification Treatments Used in Solid Foods and Feeds

In this section, we summarize the chemical, physical, and biological treatments applied separately, their effect on the reduction rates of different mycotoxins, and the quality of the treated materials.

### 2.1. Chemical Treatments

Chemical decontamination is used in many industries [26]. It can be used for the destruction of mycotoxins or their neutralization [39]. Many chemical agents are used for the decontamination of solid foods and feed, such as limewater [40], organic acids [41], ozone [42], and ammonia [43]. All these treatments are discussed below in this section (Table 1).

#### 2.1.1. Nixtamalization

Nixtamalization is a traditional process used for maize. It is a chemical treatment based on the alkaline hydrolysis of aflatoxins by the addition of lime and subsequent cooking for a predefined time. This process causes the opening of the lactone ring of aflatoxins leading to their inactivation by the high pH medium and heating process. The efficiency of this process is related to many factors, such as the quantity of the lime used, the temperature of the process, and the contact time between the solution and the grains [60]. The disadvantage of the traditional nixtamalization process (TNP) is the generation of a large quantity of wastewater and a large amount of nejayote (water containing solid fractions of maize tip cap, pericarp, germ, and aflatoxins). Nejayote imposes a safety problem because it is reused in some regions as animal feed, for another nixtamalization process, or to water plants [51].

Inconsistent results have been shown concerning the use of the traditional nixtamalization process to reduce aflatoxins in corn. High reduction rates of aflatoxin B_1_ (AFB_1_) and aflatoxicol of 96% and 70%, respectively, were achieved by Anguiano–Ruvalcaba et al., supporting the use of TNP to mitigate aflatoxins in corn [61]. Another study was conducted in the Huasteca Potosina region in the central part of Mexico to measure the efficiency of TNP and showed that this process is not efficient enough to mitigate the aflatoxins in maize grains [51].

To determine the optimal pH for the alkaline treatment of maize dough used for tortilla production, an alkaline treatment at pH 10.2 was appropriate to achieve the total elimination of aflatoxin AFB_1_ with a resting time of 30–40 min at room temperature. [52].

The cooking ingredients used to perform the nixtamalization may have a critical role in mitigating aflatoxins and decreasing the detrimental effect on food and feed quality, as well as the environment [53]. A study was conducted on maize, and two cooking solutions were used (1% slaked lime or 1% traditional liquid ash). Their efficiencies were similar, and the reduction rates of 90% and 80% of aflatoxins and fumonisins, respectively, were achieved by soaking the maize grains. The flours prepared from the treated grains showed a decreased peak viscosity, as compared with the non-nixtamalized maize flours, associated with a slight reduction in the fat, sugar, protein, and dietary fiber contents. The ash and the niacin content were increased, and the acceptability of the products produced using the treated grains by consumers was high, making this method a cost-effective alternative to fumonisins and aflatoxins detoxification of maize [50].

Another study showed that calcium hydroxide is consistently used for cooking maize grains and for producing nixtamal, but this causes environmental pollution issues due to the high pH of the wastewater and the byproducts ensuing from this process. Alternatively, the authors proposed the use of different cooking solutions, such as sodium and potassium hydroxides, that can be used as an alternative to calcium hydroxide after showing their effective reducing effect on fusarium mycotoxins [53].

#### 2.1.2. Ozonation

Ozone (O_3_) is a greenhouse gas made of three oxygen atoms. It is present naturally in the atmosphere (friend O_3_) or generated by human beings (foe O_3_) [26]. It can be produced by several methods, including UV-irradiation, electrical discharge of oxygen, and electrolysis of water. It is a highly reactive molecule having a high oxidizing effect (redox potential = 2.07 V), and it is used to mitigate many types of contaminants in food [62].

Ozone is used to reduce mycotoxins in food and feed, and it showed sanitation and antimicrobial effectiveness against viruses, bacteria, spores, and fungi [44]. The application of ozone to reduce or eliminate mycotoxins in foods and feeds can be performed by fumigation or in solution; the latter method is faster, as confirmed by a study where ozone solution at a concentration of 10 mg·L^−1^ was used to treat a 1 μg mL^−1^ of deoxynivalenol (DON) solution for 30 s achieving a degradation rate of 54.2%. Meanwhile, the degradation rate was higher in scabbed wheat (moisture content = 17%) reaching 57.3%, when treated for 12 h with ozone gas (Concentration = 60 mg L^−1^). The high degradation rate of DON can be established by increasing the concentration of ozone in solution and gas and by prolonging the application time [47].

Ozone fumigation of sun-dried herbs and spices showed that the main factors for achieving high microbial reduction and high aflatoxin degradation are the concentration of ozone and the exposure times. Ozone fumigation at 3 ppm for 210 min showed a considerable decrease in the total aflatoxins concentration by 93.75% in licorice and by 90% in peppermint. The disadvantage of this method is the reduction of the essential oil of chamomile by 57.14% and peppermint by 26.67% [44].

The degradation of trichothecene mycotoxins by aqueous ozone showed a pH sensitivity with maximum effectiveness at acidic pH of 4–6 and no effectiveness at alkaline pH of 9 [46].

Ozone treatment can be implemented during the storage of crops in a silo. In a previous study aiming at the decontamination of filamentous fungi, it was difficult to achieve homogeneity of ozone concentration in the silo during rice treatment following the application of 0.393 kg O_3_ m^−3^ rice. The highest concentration of ozone was in the lower part of the silo, at the proximity of the ozone’s inlet, suggesting a strong reduction of fungi in this area, while the effect of ozone gradually decreased while moving toward the upper parts of the silo [49].

Ozone is naturally unstable, leaving no residues in foods and feeds after its transformation into oxygen in the treated samples. It has a GRAS status (generally recognized as safe). It causes no waste and does not impose pollution problems [44,47]. At the industrial scale, the cost of ozone must be considered in developing countries [63].

Ozone gas applied to the rice silo did not damage the rice quality. No significant changes were perceived, especially in starch modifications, lipid peroxidation, protein profile, and microstructure alteration [49]. Concurrently, the treatment of parboiled rice during the maceration stage showed many advantages in the quality of the treated rice, such as higher head rice yield, higher luminosity and hardness, decreased cooking time, percentage of defective grains, and the abundance of soluble protein [45].

As shown in Table 1, the ozonation appears to be more effective in reducing the microbial load than in reducing mycotoxins already produced in food or feed. Higher aflatoxin reduction rates are achieved in powdered herbs and spices than in intact grains, wheat, and rice. The organoleptic and nutritional characteristics are affected differently in diverse food matrices. They range from no significant modification, or a slight improvement in quality attributes of wheat and rice, to a detrimental effect on the essential oil of herbs and spices [44,47,48,49].

#### 2.1.3. Ammoniation

Ammonia (NH_3_) is a gas stored in water solution or pressurized bottles. It is used to detoxify mycotoxins in different food matrices. Most studies have focused on aflatoxins [37]. Many studies supported the use of ammonia to detoxify aflatoxins in foods and feeds and proposed it as an effective, economic alternative [55,56]. High reduction rates of aflatoxins have resulted in ammoniation reaching 96 and 99% [54].

Ammonia is more effective against aflatoxins G_1_ and G_2_ than aflatoxins B_1_ and B2. This is confirmed by a study that demonstrated that the degradation rate was 95% for aflatoxin G_1_, 93% for aflatoxin G_2_, 85% for aflatoxin B_1,_ and 83% for aflatoxin B_2_ in artificially contaminated maize crops [56].

The highest efficiency of ammoniation in aflatoxin detoxification is achieved by the use of 0.5 to 2.0% ammonia at moisture levels between 12 and 16% and under pressure (45–55 psi) at high temperatures reaching 80–100 °C for 20 to 60 min where the recovery of the ammonia is conducted by evaporation at the end of the process [54,64].

The degradation of deoxynivalenol (DON) in contaminated wheat kernels was confirmed, and the achieved degradation rates were 75% or higher. The initial concentrations of DON in the kernels were up to 2000 μg/kg and vapor ammonia was implemented at a high temperature reaching 90 °C for 2 h. The toxicity of the ammoniation products is lower than DON [37].

Aflatoxins-contaminated corn is detoxified by the use of aqua-ammonia (liquid) or anhydrous ammonia (gas). The treatment with aqua-ammonia imposes the drying of the crops before storage. Ammoniation could affect the organoleptic characteristics of the treated corn by causing grain darkness as a result of the caramelization of sugar (altrose) caused by the increase in the temperature during treatment [55].

#### 2.1.4. Acid

Food-grade acids can be used for the degradation of many mycotoxins [62]. These acids affect mycotoxins differently. Ochratoxins are reduced through their conversion to phenylalanine and lactone acid. Aflatoxins could be reduced by an acid-catalyzed addition of water to the vinyl ether double bond of AFB_1_ and AFG_1,_ and they will be converted to their hemiacetal [58].

The high efficiency of food-grade acids in the reduction of aflatoxins is confirmed when used at a concentration of 9% for 15 min at two moisture levels (10 ± 3% and 16 ± 3%), reaching 99% for citric acid, 99.9% for lactic acid, and 96.07% for propionic acid. The most favorable results are obtained following the use of citric acid because of its efficiency in the reduction of the four aflatoxins (AFB_1_, AFB_2_, AFG_1_, AFG_2_) without the formation of any hazardous residues or metabolites [57].

The conversion rate of AFB_1_ to AFB_2a_ by citric acid solution (1 M) reached more than 97% when implemented at room temperature for 96 h. This rate increased to 98% and the process was accelerated so that it could be accomplished in only 20 min when boiling was used [59].

Another study evaluated the effect of citric acid and lactic acid solution on the reduction of DON and its derivatives. This study showed that the 5% solutions of both acids are effective in reducing the DON and its derivative 15Ac-DON but have no or small effect on zearalenone, fumonisins, and culmorin [58]. The organic acid may affect the quality of some products causing discoloration and slight changes in odor and taste [59]. The high cost of organic acids is a challenge for their use in the detoxification of feeds [62].

The instability of ozone may support its safety in being used for the degradation of mycotoxins in food and feed without leaving harmful residues in the treated materials. The results shown from different studies in Table 1 put forward the efficiency of the ozonation and ammoniation. Ammonia was able to achieve higher reduction rates of AF and DON than ozone in different food materials. It is worth noting that ammoniation has not been approved by the FDA and may cause many sensorial quality problems [55], and its effect on aflatoxins may be reversible in an acid medium such as the gastrointestinal tract [54]. Many studies proved the high efficiency of citric acid solution in the reduction of mycotoxins and, especially, aflatoxins, reaching 99% with few detrimental effects on food quality. The limitation of its scalability and its use at the industrial level is related to the high cost of these acids [58].

Nixtamalization is a processing step of maize that can be used as a means to eliminate or reduce the number of mycotoxins [65]. Contradictory results about the effectiveness of the traditional nixtamalization process (TNP) are presented in Table 1. Maureen et al., proposed the high efficiency of this process to reduce AF (up to 90%) [50], while Rodríguez–Aguilar et al., declared that TNP is not efficient in reducing the AF in maize and confirmed the contribution of this process to environmental pollution through the high amount of wastewater generated during its execution [51,52]. A potential solution to this harmful effect on the environment could be attributed to the change in the cooking ingredient [53]. The absence of the ideal chemical treatment for the mitigation of mycotoxins imposed the necessity to find other alternatives to be discussed in the next section.

### 2.2. Physical Treatments

Many traditional methods, such as cleaning and sorting, are used. These methods are capable of physical separation by removing the contaminated portions from the crops and preventing the transfer of the pathogens to the non-contaminated portions. These are not able to neutralize or degrade the mycotoxins already produced in the crops; they only isolate the contaminated portions [28,66,67]. Several industrial processes require the use of conventional cooking at temperatures below 100 °C. Most mycotoxins are heat stable, and it is not possible to mitigate them by these conventional heat treatment processes [68].

The physical detoxification methods explained in this section (Table 2) include thermal treatments or invasive methods such as extrusion [68], and non-thermal treatments or non-invasive methods such as photocatalysis [69], cold plasma [70,71], electrolyzed oxidizing water [72], and irradiation [73,74]. These technologies are beneficial since they are safely used for many food matrices without causing negative effects on the nutritional and organoleptic quality of treated food. The limited scalability at the industrial level may be considered a disadvantage of many physical treatments [75].

#### 2.2.1. Photocatalytic Treatment

The use of UV-visible irradiation combined with a semi-conductive photocatalyst showed high efficiency in the reduction of aflatoxins in a liquid medium [62]. DON was degraded in contaminated wheat samples by 72.8% following the use of photocatalyst UCNP@TiO_2_ (8 mg mL^−1^) for 90 min with a ratio of wheat to liquid of 1:2 [76].

The photocatalytic efficiency NaYF_4_:Yb,Tm@TiO_2_ on the degradation of DON was greater in solution than in wheat. This decrease in efficiency may be caused by the attachment of toxins to starch or proteins in the wheat, or to other wheat components, or even the shielding effect of wheat grains that hinder the light from reaching all contaminated surfaces. A complete degradation was achieved in a solution containing 10 μg mL^−1^ of DON when treated with simulated sunlight using NaYF_4_:Yb,Tm@TiO_2_ (6 mg mL^−1^) at pH 7 for 60 min. This rate was decreased to 69.8% when artificially contaminated wheat was treated with UCNPs aqueous solution (Ratio 1:1) by illuminating the samples with a Xe lamp for 120 min [69].

Many factors make the use of these techniques more advantageous. They are completely inorganic and do not result in the formation of secondary metabolites, causing pollution. They are cost-effective and easy to apply, requiring mild conditions [95]. Photocatalysis did not cause significant changes in the starch, the protein contents, or the amino acid value in wheat. However, there are also many disadvantages, such as the increased yellowness and decreased whiteness of the wheat flour, the damaged surfaces of starch granules when a prolonged illumination time is applied, and decreased fatty acid value, wet gluten content, and pasting properties of wheat [76].

#### 2.2.2. Cold Plasma

The common states of matter are solid, liquid, and gas; plasma, the fourth, uncommon state, is formed by supplying enough energy to substances to assure the transition from the solid to the ionized state [96].

Cold atmospheric plasma (CAP) is a non-thermal technology that has shown its efficiency in reducing fungal pathogens and their toxins [64]. This technology is a successful alternative to the traditional treatments (heat treatment, wet chemistry, or UV-irradiation) usually implemented. These treatments proved their inefficiency in mitigating AF without affecting the food and feed quality [77].

Complete degradation of aflatoxin B1 is achieved by the effect of CAP-RONS (Reactive Oxygen Plasma and Nitrogen Species), which have a high oxidative potential and high affinity to react with the vinyl bonds in organic molecules. These RONS are yielded by the generation of non-equilibrium atmospheric in ambient air. The same degradation rate is achieved by applying the same CAP system to contaminated corn kernels. CAP treatment is faster than UV-C treatment and is significantly more efficient in AFB_1_ reduction than UV-C treatment [77].

A different study explored the effect of low-pressure dielectric barrier discharge (DBD) plasma on the degradation of T-2 and HT-2 toxins in oat flour by using different working gases. Only partial degradation of T-2 and HT-2 toxins was achieved by applying this technology, and all the experiments yielded similar results to those obtained by the thermal treatments usually applied during food processing, such as cooking, extrusion, or roasting. None of the used gases in this study could completely detoxify the oat flour samples. The highest degradation rate of T-2 and HT-2 toxins reached 43.25% and 29.23%, respectively, after treating the samples with nitrogen for 30 min. CAP treatments using molecular oxygen and air as working gases did not affect these toxins [78].

Another study confirmed the results of the previous one that DBD plasma is not capable of achieving complete detoxification of AFB_1_ and FB1 in food matrices. In this study, spiked maize kernels were exposed to a pulsed dielectric barrier discharge (DBD) plasma jet for 10 min. The detoxification rate was 65% for maize grains spiked with AFB_1_ with an initial concentration of 1.25 ng/g and 64% for maize grains spiked with FB1 with an initial concentration of 259 ng/g [79].

Roasted coffee beans artificially contaminated with ochratoxin A (OTA) were treated with cold plasma for 30 min, and the degradation rate reached 50%. This result was satisfactory as per the EU standards. The brine shrimp lethality assay was used to evaluate toxicity, and the result was “Toxic” for the untreated beans and “Slightly Toxic” for the treated ones [80].

CAP has a low detrimental effect on the organoleptic and nutritional quality of foods and feeds. This treatment could explain this treatment’s low penetration depth, so the degradation may affect only the superficial layers and protect all internal components. The generated RONS may affect the antioxidants and lipids present in the product [79]. SBD plasma has been shown to be more efficient than DBD plasma in the reduction of aflatoxins and other mycotoxins (Table 2), which seems to be more practical and scalable at the agri-food industrial level [77].

#### 2.2.3. Pulsed Light

Pulsed light is a non-thermal treatment used to improve food safety and maintain the quality of food products by preventing the effect of heat treatment adopted in other techniques. It is generated by the flash repetition of non-coherent, broad-spectrum, high-intensity light [97]. It includes infrared, ultraviolet, and visible rays. It has been FDA approved since 1996 to be used for the superficial decontamination of food products (maximum fluence 12 J cm^−2^) [62,98].

This technology achieved higher reduction rates of AFB_1_ and AFB_2_ in rice bran than in rough rice because of its high efficiency on the surface and external parts [83]. Another study showed a positive relationship between the aflatoxins degradation rate and the initial concentrations in solid medium and the intensity of pulsed light treatment [82].

The effect of pulsed light on red pepper powder to mitigate microorganisms and mycotoxins such as total AF, AFB_1_, and OTA was investigated. The application of 61 pulses at high fluence (9.1 J/cm^2^) for 20 s effectively reduced the yeast and molds and the total plate count in red pepper powder. The same treatment parameters were applied to cause the reduction of total aflatoxins by 50.9%, aflatoxin B_1_ by 67.2%, and ochratoxin A by 36.9%. Total phenols increased apparently and significantly, while the total color was slightly changed [81].

#### 2.2.4. UV-C Irradiation

UV light is another non-thermal treatment used as an alternative to thermal and chemical treatments to reduce the negative effects on the quality of treated foods and prevent the formation of residues and byproducts. UV light is classified into different bands according to the wavelength used. UV-C, which ranges from 200 to 280 nm, is commonly used because of its high efficiency against microorganisms, specifically at 250 and 260 nm [99].

The effect of UV-C irradiation on different types of rice was studied using UV irradiation at 254 nm for 1 and 3 h (moisture content of rice = 13%). The one-hour treatment was able to achieve a dose of 2.06 KJ cm^−2^, causing fungal decontamination and mycotoxin reduction in black and red rice without affecting the cooking and color characteristics. In contrast, the three-hour treatment increased the dose to 6.18 KJ/cm^2^ and increased the efficiency with a reduction of the total phenolic compounds. In brown rice, only the high dose achieved by the three-hour treatment was effective in reducing the fungal decontamination while causing undesirable browning of grains [73].

Low penetrability and the shadowing effect are two hurdles to the success of UV-C irradiation [100]. To overcome these problems and to increase the efficiency of UV irradiation, a customized rotational cylindrical chamber was established by Shen and Singh. In this study, the authors used a UV indicator applied to peanut kernels and treated with UV-C irradiation at 2.3 mW cm^−2^ for 2 h with a continuous rotational movement at 11 rpm. The uniformity of the UV-C treatment was significantly improved when the reduction percentage of AFB_1_ was increased by 23.4% [85].

In another study, innovative vibrational decontamination equipment was designed for the decontamination of maize and peanut to increase the efficiency of this technology. UV-C irradiation was applied at a range of 1080 to 8370 mJ cm^−2^. After incubation for 10 days, the samples irradiated with 8370 mJ cm^−2^ showed the lowest count of *A. flavus* in peanuts and maize. AFB_1_ reduction rates reached 43% and 51% for maize and peanut, respectively [84].

#### 2.2.5. Gamma Irradiation

Gamma irradiation is a treatment that can be used to disinfect crops by reducing the number of fungi or by mitigating mycotoxins already produced by the fungi in these crops [64]. A gamma source, such as cobalt-60, must be used to generate very high-energy photons. These photons are capable of killing spoilage and pathogenic microorganisms by causing damage to their DNA. The free radicals and ions that occur after the interaction of the energy with water molecules present naturally in food products or crops will attack microbial DNA [101,102].

The important role of water in the successful use of gamma irradiation was supported by a study that investigated the degradation rate of OTA in aqueous solution and different food products (wine, grape juice, and wheat flour). The sensitivity of OTA irradiated at 30.5 kGy reached the maximum in water solutions. It was also demonstrated that OTA is highly resistant to the same irradiation dose in solid matrices or dry foods [87].

Another study confirmed the use of gamma irradiation to inhibit *A. flavus* and *A. ochraceus* and reduce AF and OTA in maize. In this study, low doses of 6 kGy lead to the inhibition of mold growth. The reduction of the formed AFB_1_ by 40.1%, AFB_2_ by 33.3%, and OTA by 61.1% in maize required higher doses (20 kGy) [86]. A different gamma irradiation study was performed on sorghum and showed that the reduction of natural fungi in sorghum reached 90% at 3 kGy with maximum reduction rates of 59% for AFB_1_ and 32% for OTA realized at 10 kGy [88].

#### 2.2.6. Extrusion

Mycotoxin reduction could result from food processing operations, such as extrusion, which can simultaneously improve product quality and increase food safety levels by reducing toxins [103]. Conventional cooking treatments (conducted at temperatures below 100 °C) cannot participate in the mitigation of mycotoxins in food products because most of these toxins are heat stable. Alternative cooking treatments, such as extrusion, are performed at higher temperatures and show efficiency in reducing mycotoxin contamination [68].

A study conducted by Massarolo et al. used a single-screw extruder at 50 ng g^−1^ to reduce aflatoxins on spiked cornmeal samples. The reduction rates of all aflatoxins were higher in the samples after the addition of high amylose corn starch, reaching 89.9% for AFB_1_ and AFG_2_, 88.6% for AFB_2_, and 75% for AFG_1_. Extrusion may cause possible interactions of the toxins with food components, decreasing their bio-accessibility. Their availability in the small intestine increased significantly after digestion [90]. Another study was conducted by Janić Hajnal et al., and focused on the effect of co-rotating twin-screw extruder on other mycotoxins (DON, 3- and 15-AcDON, HT-2, TEN, and AME) in whole grain triticale flour. The optimal reduction rate of all studied mycotoxins was achieved at a screw speed of 650 rpm with a feed rate of 30 kg/h and moisture content of 20 g/100 g. A higher reduction rate was found in AME, while the lowest rate was detected for DON [89].

#### 2.2.7. Electrolyzed Oxidizing Water

EOW is prepared by introducing tap water and salt into an electrolysis chamber. It is considered a sanitizer or disinfectant because of its bactericidal and fungicidal effects. It is characterized by its specific pH value, its oxidation-reduction potential, ORP, and the available chlorine concentration, ACC [104]. The physicochemical properties of EOW used to treat foods are ACC from 10 to 100 ppm and ORP from −800 mV to higher than 1000 mV. The pH depends on the type of water used in the research, acid, slightly acid, neutral, or alkaline electrolyzed water [104,105]. Research was conducted to study the effect of different pH values of EOW on fungal elimination and DON reduction in wheat grains. This study showed that for acid-electrolyzed water, the optimal pH for reducing fungi was 2.5 and 5.5 to eliminate DON. For alkaline electrolyzed water, the optimal pH value to eliminate DON was 9.5, while pH values between 8.5 and 12.5 were effective for eliminating fungi also. The optimal pH values of the alkaline EOW (pH 9.5) and the acid EOW (pH 5.5) did not affect the wheat characteristics such as color, moisture content, and protein and gluten contents. The starch morphology also did not change significantly. A beneficial effect was caused by the acid-electrolyzed water on wheat flour, causing higher stability, increased farinograph quality numbers, and lowered softening degree [91].

#### 2.2.8. Electron Beam Irradiation

The electron beam is a type of ionizing irradiation [99]. It is generated by the use of a safe dose of an electric accelerator [63]. It is a non-invasive, non-thermal, and eco-friendly detoxification method used for cereal-based products in order to reduce microbial and mycotoxin contamination [106,107]. This irradiation treatment was used to decontaminate naturally contaminated red pepper powder. Low doses of 6 and 10 kGy reduced the yeasts and mold count and the total plate counts by 3, 4.4, and 4.5 log CFU/g, respectively. A higher dose of 30 kGy achieved a 25% reduction in OTA. Electron beam irradiation is more effective in the reduction of microorganisms than mycotoxins. It is worth noting that this treatment had low detrimental effects on the quality of the treated pepper powder, causing a slight change in color and less than 15% reduction of total phenols, carotenoids, and antioxidant activity [92]. Another study conducted by *Kim et al*. used irradiation as a non-thermal decontamination method for an uncooked Korean cereal product called Saengshik. Electron beam irradiation was conducted at 10 kGy and showed an increase in the total phenolics and a decrease in the total carotenoids and chlorophylls with preservation of antioxidant capacity when the irradiation was conducted at doses lower than 10 kGy [108].

#### 2.2.9. Milling

The milling process is effective in reducing mycotoxins in feeds and foods [68,109]. The weakness of this method lies in the redistribution of mycotoxins in the resulting fractions of milling and their concentration in the products intended for animal feed [67].

Scarpino et al., showed that cleaning maize grains may cause a reduction in the fungal metabolites by 1.2 to 2 times. In this study, the milling process of maize kernel caused an unequal redistribution of the mycotoxins in the different maize fractions and concentrated most mycotoxins in the germ. The highest mycotoxin contents were found in animal feed products, and the healthier products are large flaking grits [93]. A study was conducted using this principle by implementing dry and wet de-germination to maize and showed that the latter was more efficient for decreasing fumonisins in the milled products. Cleaning the kernels reduced FBs by 42%. Furthermore, the tempering degermination process of the uncleaned kernels achieved high reduction rates as compared to the dry degermination, reaching 94% for the largest-sized flaking grits. This process was able to facilitate the separation of the horny endosperm from the fine milling fractions. [94].

By evaluating the results of the different chemical treatments in Table 1 and the results of the different physical treatments in Table 2, we can conclude that chemical treatments are able to achieve the highest degradation rates of different mycotoxins in solid foods and feeds. In contrast, the physical treatments achieve lower degradation rates, but their effects on the quality of treated materials are smaller.

The shadowing or shielding effect is the principal limitation related to the use of photocatalysis and UV-C irradiation in the reduction protocols of mycotoxins in solid food materials. Many studies (Table 2) tried to overcome this limitation by rotating the irradiated peanuts to ensure UV uniformity, but no significant increase in the reduction rates occurred. Photocatalysis achieved higher reduction rates of DON, reaching a total elimination of DON in wheat [76]. CAP is a superficial treatment with low penetration depth. It showed good efficiency in reducing AFT and AFB_1_ without deterioration in the quality of the treated product. Furthermore, SBD plasma was more effective than DBD plasma since it achieved the complete elimination of AFB_1_ [77,79]. As working gas, nitrogen achieved the highest reduction rates as compared to oxygen and air when using the DBD system [78]. The pulsed light effectiveness was superficial and showed greater AF reduction rates when applied to rice bran than to rough rice. The AFB reduction rates are defined by the PL intensity and the initial concentration of mycotoxins in the food to be treated [83]. Gamma irradiation showed good effectiveness in reducing mycotoxins in food containing a high amount of water and reducing the fungal load in solid food. Low gamma irradiation doses were able to eliminate fungi and reduce mycotoxin formation in maize, but higher doses were required to reduce the already produced OTA in this material. Mycotoxins were not completely eliminated in any of the mentioned studies (Table 2) [86,87,88]. The electron beam showed its efficiency as a disinfectant by the reduction of different microorganisms such as bacteria, yeasts, and molds, but it was not effective in the reduction of OTA in red pepper powder [92]. The milling process resulted in the reduction of mycotoxins in many edible fractions of maize but caused the concentration of these fungal metabolites in the germ, especially in the fractions used as animal feeds [93].

### 2.3. Biological Treatments

Biocontrol showed high efficiency in the prevention of AFs formation in the pre-harvest stage when non-aflatoxigenic biological control strains are inoculated in the fields and competed with aflatoxinenic strains of *Aspergillus* for nutrients and place and causing their exclusion [110,111]. The studies discussed in this section aimed to mitigate the already formed mycotoxins in feeds and foods by biological treatments and not to prevent their formation in crops (Table 3).

Most studies about the mitigation of mycotoxins by biological means focused on the treatment of liquid food or milk [32,33,112], assessing the effect of yeast, bacteria, or their enzymes on the mycotoxins in buffers or solutions [30,113,114]. Biological detoxification could be the result of binding the targets by adsorption mechanisms or by degradation. This detoxification of mycotoxins can be conducted using microorganisms (bacteria, biofilm, or yeast) or their metabolites and enzymes [115]. In this section, we screen various studies using biological control strategies to mitigate the mycotoxins in solid food and feeds (Table 3).

ZEN-detoxifying *bacillus* strains were used to detoxify highly contaminated maize with an initial concentration of 5 mg kg^−1^ of ZEN. The degradation of ZEN is related directly to the esterase activity, which has been found in all tested strains, with the maximum activity in B1 and B2 strains. The highest ZEN degradation rate was attained in B2 strains, reaching 56%. B2 strains showed their efficiency in the detoxification of other mycotoxins with different rates—AFB_1_: 3.8%, DON: 25%, FB1: 39.5%, T2 toxin: 9.5%. The presence of ZEN enhanced the fermentation process of the contaminated maize compared to the non-contaminated grains [116].

CotA laccase is found in the endospore coat of *Bacillus*. It protects spores from UV light and hydrogen peroxide and has an oxidizing capacity. CotA laccase was immobilized onto chitosan microspheres and used to degrade ZEN in artificially contaminated cornmeal samples. The free CotA laccase form achieved a degradation rate of 70%, while the immobilized form was faster and more effective, achieving a higher degradation rate reaching 90%. The most important advantage is the reuse of the immobilized enzyme. Guo et al., showed that the degradation rate decreased to 54% following multiple uses of the immobilized CotA laccase in the third cycle, reaching only 21% in the fifth one [117]. Lactic acid bacteria (LAB) were used to mitigate mycotoxins in wheat-based products. The *Pediococcus acidilactici* LUHS29 strain achieved the highest reduction rates of mycotoxins when used alone in sourdough fermentation for 48 h. It removed 15-AcDON, AOH, D3G, toxins H-2 and HT-2, completely removed ENNB1, and reduced the DON by 44–69%. The combined fermentation using this LAB with *Lactobacillus Plantarum* LUHS135 strain showed great efficiency and increased the reduction rate of DON to 79–100% [118]. In a study conducted by Alberts et al., enzymatic detoxification was examined using Fumonisin Esterase FumD to degrade FB in maize. This enzyme can hydrolyze and remove the tricarballylic acid groups when added to maize during the conditioning step (for 250 min) during the dry milling process. The use of 40 U/kg of FumD in maize resulted in a 99% degradation of FBT in total hominy feed but did not accomplish any degradation of FBT in super maize meal [119].

The fungal growth and/or the mycotoxin production was controlled in bread using specific yeast strains and achieving reduction rates varying between 16.4 and 33.4% for DON, 18.5 and 36.2% for NIV, and 14.3 and 35.4% for ZEA [120]. The heat treatment of peanut samples at 100 °C for 15 min before solid-state fermentation by *Zygosaccharomyces rouxii* showed great efficiency in the mitigation of AFB_1,_ and the reduction rate reached 97.52% [121].

**Table 3 foods-11-03304-t003:** Microbial and enzymatic treatments for the reduction of mycotoxins in solid foods and feeds.

Treatment	Feeds/Foods	Contaminants	Experimental Parameters	Reduction Rates	Advantages	References
Bacteria: ZEN-detoxifying *Bacillus* (ZDB) strains	Maize	ZEN	The highest level of ZEN degradation	B2 strain-reduction rate = 56%	Esterase activity is demonstrated in all strains; The stronger esterase activity: B1 and B2 strains;	[116]
			B2 strain detoxifies other mycotoxins	Reduction rates: AFB_1_: 3.8%; DON: 25%; FB1: 39.5%; T2 toxin: 9.5%	Fermentation of ZEN-contaminated maize by B2 strain compared to ZEN-free maize: Better fermentation characteristics: (lactic acid > 110 mmol·L−1; acetic acid < 20 mmol·L−1; pH < 4.5).	
Bacteria: *Bacillus licheniformis* spore CotA laccaseapplication of immobilized laccase in contaminated corn meal	Corn meal	ZEN	Treatment with immobilized CotA laccase onto chitosan microspheres for 12-h	Degradation rate: 90%	Immobilized CotA laccase is much faster and more effective than free CotA laccase in degrading ZEN; Immobilization has higher thermal stability over free CotA laccase, maintaining about 87% of its initial activity after heat treatment at 80 °C for 30 min; Reusability: Immobilized CotA laccase could be recovered from corn meal solution and repeatedly used.	[117]
			Treatment with free CotA laccase for 12-h	Degradation rate: 70%		
			Reuse of immobilized enzymes for 5 cycles	Decreased degradation rate on each after each cycle: Cycle 1: 90%; Cycle 2: 77%; Cycle 3: 54%; Cycle 4: 30%; Cycle 5: 21%		
Bacteria—Fermentation: Lactic acid bacteria	Wheat-based products	DON 15 -AcDON AOH D3G, toxins H-2 and HT-2: Enniatin ENNB1	*Pediococcus acidilactici* LUHS29 strain	The strongest mycotoxins decontamination effect	*Pediococcus acidilactici* LUHS29 strain has the strongest mycotoxins decontamination effect; Combined fermentation showed more efficiency and complete elimination of DON.	[118]
			Prolonged fermentation at 35 °C for 48 h with *Pediococcus acidilactici* LUHS29 strain	DON: 44–69% 15-AcDON, AOH, D3G, toxins H-2 and HT-2: Removal Enniatin: 5–70% ENNB1: complete removal		
			Combined fermentation (Lactic acid bacteria 7 (JCM 1149) and Pediococcus acidilactici LUHS29 (DSM 20284))	Complete elimination or effective reduction of DON: 79–100%		
Enzyme	Maize	FB	FB degradation during dry milling of maize		Highest enzyme concentration: 32 U/100 g maize: Complete conversion into HFB1; Cost-effectiveness of upscaling the FumD FB dry milling method to an industrial level requiring up to 40,000 U FumD/ton maize, will depend on the safety benefits of consuming the milling products as well as the commercial value of the total hominy feed lacking FBT.	[119]
Fumonisin esterase FumD			Enzyme concentration: 40 U/kg	Reduction rates FBT: 99% in total hominy feed; 48% in semolina; 7% in special maize meal No reduction in super maize meal.		
Yeast	Wheat grains and bread	Fusarium Mycotoxins: DON, NIV ZEN	Bread prepared by baking with the addition of an inoculum of the test yeast	Reduction rates: DON: 16.4% to 33.4%; NIV:18.5% to 36.2%; ZEA: 14.3% to 35.4%	The biocontrol yeasts strains may arrest fungal growth, reduce mycotoxin production, or both.	[120]
Yeast	Peanut meal	AFB_1_	Peanut samples are heated at 40, 60, 80, 100, or 110 °C for 10 min			[121]
			The residual rates after heat treatment at the following temperature for 10 min: (T:% of residual AFB_1_	80 °C: 61.08%; 100 °C: 63.46%; 110 °C: 49.63%		
			The residual rates after fermentation by *Z. rouxii*: (Temperature: % of residual AFB_1_)	(40 °C:32.73%)-(60 °C:20.85%)-(80 °C:16.18%)-(100 °C:5.13%)-(110 °C:5.10%)		
			100 °C	The optimal temperature achieved the highest reduction rate		
			Peanut samples are heated at 100 °C for 5, 10, 15, or 20 min			
			The residual rates after heating at 100 °C for different times: (time: % of residual AFB_1_)	(5 min: 21.06%)-(10 min: 5.13%)-(15 min: 2.48%)-(20 min: 2.44%)		
			15 min	The optimal time		
			Optimal treatment (100 °C -15 min):	Residual % of AFB_1_: 2.48%		

## 3. Subsequent Detoxification Treatments Used in Solid Foods and Feeds

The efficiency of the above-mentioned techniques on mycotoxin reduction showed great variability when implemented singly [122]. The additive or synergistic effect of using many combined treatments subsequently are summarized in Table 4 and evaluated below.

### 3.1. O_3_, UV-C, and Citric Acid

Ozone and acid treatment, when implemented individually, can reduce AFB_1_ and AFG_1_ more than AFB_2_ and AFG_2_. In contrast to O_3_ and acid treatment, UV-C by itself has great efficiency in the degradation of AFB_2_ and AFG_2_. This difference made the combination of the three treatments a great opportunity to increase the degradation rates of aflatoxins in contaminated pistachio samples. The subsequent treatments of the contaminated pistachio with 3N citric acid, followed by O_3_ exposure for half an hour, and UV-C irradiation for 36 h, achieved high reduction rates of more than 90% for AFB_1_ and AFB_2_ and more than 99% for AFG_1_ and AFG_2_. This combination did not cause significant changes in the organoleptic and nutritional quality of the pistachio compared to non-treated pistachio samples [123].

### 3.2. Extrusion and Fermentation

Extrusion is a type of high-temperature treatment, and as discussed previously, it can decrease the number of mycotoxins in cereals [68]. Contradictory results were indicated by Zokaityte et al. They found that extrusion may affect the mycotoxin levels differently by increasing, decreasing, or not changing their concentrations in the samples. The combination of extrusion at different temperatures (115 and 130 °C), over different screw speeds (16, 20, and 25 rpm), with fermentation for 24 h at 30 °C by using 2 strains of LAB (*Lactobacillus casei* and *Lactobacillus paracasei*) and their effects have also been studied. This combination increased the amount of lactic acid and decreased bacterial contamination as a result of pH reduction. The effect of extrusion on different mycotoxins contradicted the results obtained in other studies. The 15-DON concentration increased in all extruded samples, and the fermentation of the samples decreased them to acceptable levels. The capacity of fermentation to decrease mycotoxin levels in the food or feed samples may be caused by the binding capacity of LAB. Mycotoxin types and their initial concentrations in the food matrix, the physicochemical characteristics of this matrix, and the fermentation variables, such as temperature and duration, play a major role in determining the binding percentages [124].

### 3.3. Roasting and Brewing

The combination of roasting and brewing of naturally contaminated coffee beans by using the traditional Qatari method was studied to show its effect on the reduction of AF and OTA. The roasting temperature is the main factor affecting the reduction rates of AF and OTA in the coffee beans. The reduction rates were proportional to the roasting temperature. The maximum reduction rates achieved with the high roasting scheme were 61.52% and 57.43% for AFs and OTA, respectively. Brewing alone was effective in reducing OTA more than AF. Brewing showed high efficiency in the reduction of both mycotoxins in roasted coffee beans by a low roast scheme. The best combination was defined at a high roast scheme with traditional brewing, and the cumulative reduction rates were 62.38% for AFs and 64.7% for OTA. It is worth noting that roasting temperatures applied to coffee beans in Arab countries are lower than those applied in other countries to preserve the traditional organoleptic characteristics such as color and flavor [125].

### 3.4. PEF and Thermal Treatment

The effect of the thermal process, the pulsed electric field, and the combination of both treatments on the reduction of AFT and AFB_1_ were studied. Following the optimization of both treatment modalities, mycotoxins were affected by the thermal process time at alkaline pH, the thermal process temperature at neutral, and acid pH values when this process was implemented individually. The highest reduction rates were obtained after treatment at 110.36 °C for 15 min at pH 10, reaching 96.696% and 95.473% for AFB_1_ and AFT, respectively. On the other hand, PEF treatment was also optimized, and the highest reduction rates were achieved at a pulse width of 65 μs and output voltage of 26%. It seems that the combination of both treatments did not achieve a great improvement in the reduction rates of AFT and AFB_1_. As compared with the optimal thermal treatment implemented alone, this combination increased the reduction rates by 0.185% for AFB_1_ and 0.248% for AFT [126].

### 3.5. H_2_O_2_ Treatment at Moderate Temperature after Roasting

The effect of H_2_O_2_ on aflatoxins reduction in peanuts was investigated and showed higher efficiency following its application at 50 °C instead of room temperature (20 °C), while the reduction rate increased from 30% to 73%. The same H_2_O_2_ treatment (30 g/hg H_2_O_2_ at 50 °C) was implemented on unroasted peanuts for 8 h, achieving a higher AF reduction rate of 86%. The combination of this treatment with pre-roasting the peanuts at 140 °C for 10 min caused the inactivation of catalase and increased the reduction rate slightly to reach 90%. The constructive points of this combination were the preservation of the oil quality of the treated peanuts, the absence of significant weight loss, and the conservation of the peanut’s form since the temperature did not reach that of starch gelatinization. Moreover, the combination is eco-friendly, leaving no H_2_O_2_ residues after air drying the treated peanuts at 35 °C for 12 h [127].

## 4. Simultaneous Detoxification Treatments Used in Solid Foods and Feeds

In this section, we discuss many combined treatments applied simultaneously to food or feed matrices, and these combinations are represented in Table 5.

### 4.1. UV with H_2_O_2_

UV-C is used in combination with H_2_O_2_ to degrade aflatoxins in peanuts. It represents an eco-friendly technique leaving no toxic or harmful byproducts, with unique residual compounds limited to water and oxygen. The simultaneous application of these two treatments for 1 h (UV-C: 2.76 mW/cm^2^—H_2_O_2:_ 1 g/hg) accelerated the degradation rates of AF in both whole peanut kernels and milled kernels to 30% and 60%, respectively. The advanced oxidative processes (UV and H_2_O_2_) affected the quality of the oil in milled kernels and caused the darkening of the whole kernels [128].

### 4.2. Pulsed Light with Citric Acid

The combination of pulsed light with citric acid showed great efficiency on AFT, AFB_1_, and AFB_2_ in peanuts, with reduction rates reaching about 98.2%, 98.9%, and 98.1%, respectively. The chemical quality did not show significant changes as a result of this combined effect, but a significant change in the color of peanuts occurred [129]. In previously discussed studies about pulsed light treatments, the reduction rates of AFT, AFB_1_, and AFB_2_ were in the range of 39.2 to 90.2% in red pepper powder and rice with different fluence ranges applied [81,83]. Therefore, combining an acid with pulsed light can be considered a beneficial combination, demonstrating higher efficiency in mitigating aflatoxins than each treatment alone.

### 4.3. Infrared with Alkaline Treatment

There are contradictory results concerning the nixtamalization efficiency in reducing aflatoxins in maize by using the traditional process. Rodríguez–Aguilar et al., proposed the non-efficiency of the traditional nixtamalization process (TNP) in the elimination of aflatoxins from contaminated maize [51]. Meanwhile, Zavala–Franco et al., found that TNP can degrade aflatoxins in maize by 98.35%. The same study proposed using infrared as an alternative to heat treatment in the nixtamalization process. The applied protocol satisfactorily achieved a degradation rate of 93.82%. No formation of AFB_1_-Lys occurred with the infrared nixtamalization process. This combination of alkaline treatment with infrared seems to be promising for mitigating mycotoxins while generating fewer toxic materials than the traditional process [130].

### 4.4. Roasting with Acid

Aflatoxin levels can be reduced by using high-temperature treatment, such as roasting. Degradation rates in the range of 50 to 70% in peanuts and the range of 40 to 80% in maize were achieved [68]. Roasting pistachio nuts at 120 °C for 1 h is optimized when used in combination with the addition of citric acid and lemon juice. The high amount of used acids increased the AFB_1_ degradation rate to reach 93.1%, which negatively affected the physical quality of the pistachio. Decreasing the acid amount by half decreased the degradation rate to 49.2% but maintained the desired appearance of the treated pistachio [131]. 

## 5. Comparison between the Different Mycotoxin Decontamination Treatments

In general, chemical treatments achieved higher reduction rates of mycotoxins than physical treatments in solid foods and feeds. This effectiveness is accompanied by many side effects, such as the detrimental impacts on the quality of the treated food materials (ammoniation) and the formation of unavoidable chemical residues causing an environmental problem (nixtamalization). All the chemical treatments presented in this review showed possible scalability, except the acid treatment, due to the high cost of using organic acids. The physical treatments showed lower degradation rates. This can be seen in the shielding effect in the case of irradiation or the presence of the skin on some foods, such as peanuts. It is worth noting that these physical treatments usually have low penetrability where the effect remains superficial, treating a thin layer. The biological treatments of solid foods or feeds were commonly less available than other treatments. They showed good results and achieved high reduction rates with a beneficial effect of LAB on fermentation by increasing lactic acid production in maize. Combined fermentation using two LAB strains achieved higher reduction rates than those using each strain individually. The results obtained using the biological decontamination treatments prove its suitability to be considered an alternative to physical and chemical treatments by providing a safe, eco-friendly, and cost-effective method with a minimal negative effect on the quality of treated materials.

Concerning the combined treatment and by comparing the subsequent treatments in Table 4 and the simultaneous treatments in Table 5, we can spotlight many successful combinations, such as the subsequent application of O_3_/UV-C/citric acid and high concentration H_2_O_2_ treatment at moderate temperature/roasting, which achieved a reduction of AFs in pistachio and peanuts, respectively. The reduction of AFs by PEF/heat treatment attained high reduction rates in agar, but it was decreased when implemented in dry food, hypothesizing that the presence of water contributes to its success in AF elimination. Roasting/brewing was able to reduce mycotoxins without reaching the complete elimination of AFs and OTA from coffee beans. All the simultaneous treatments mentioned in this review showed their success in reducing or eliminating AFs; reduction rates exceeding 93% were accomplished by implementing citric acid with pulsed light to peanuts, IR nixtamalization to maize, and roasting with acid to pistachio.

## 6. Conclusions

In this review, we screened, evaluated, and discussed different chemical, physical, biological, and combined techniques to mitigate mycotoxins in solid foods and feeds. Many chemical treatments showed their effectiveness by achieving approximately a total elimination of AFs under certain conditions, such as optimized nixtamalization, ammoniation, and acid. Physical treatments such as photocatalysis and cold plasma were able to achieve the complete elimination of DON and AFB_1_, respectively. Chemical treatments showed higher reduction rates of mycotoxins than physical treatments, but the latter treatments were favorable from a quality perspective. Their effect was superficial, causing minimal changes in the quality of treated materials. Biological treatments are considered safe, eco-friendly, and cost-effective methods for mitigating mycotoxins. *Zygosaccharomyces rouxii* or a combination of LAB strains attained high reduction rates. Two or more treatments were used subsequently or simultaneously in order to find a synergistic effect of the combination to achieve high reduction rates in solid food materials without inducing any extreme impacts in each one. Nine combinations are presented in the last two tables, showing the higher reduction rates of aflatoxins (>90%) achieved by the following two combinations when implemented subsequently: O_3_/UV-C/citric acid and high H_2_O_2_ concentration treatment at moderate temperature/roasting. Other combinations were applied simultaneously, also showing their efficiency in the reduction of aflatoxins (>93%), such as citric acid with pulsed light and roasting with acid. These combinations affected the physical characteristics of the treated nuts. Future research should focus on the optimization of physical treatments to increase their mycotoxin reduction efficiency and on the elaboration of more combination possibilities to find the best synergistic effect to protect the product quality and the environment. It is highly important to focus more on implementing detoxification techniques on naturally contaminated materials than spiked or artificially contaminated materials. Natural contamination may be caused by several mycotoxins and may occur differently. Furthermore, these studies should examine the effectiveness of these techniques at the industrial scale more than at the laboratory scale.

## Figures and Tables

**Figure 1 foods-11-03304-f001:**
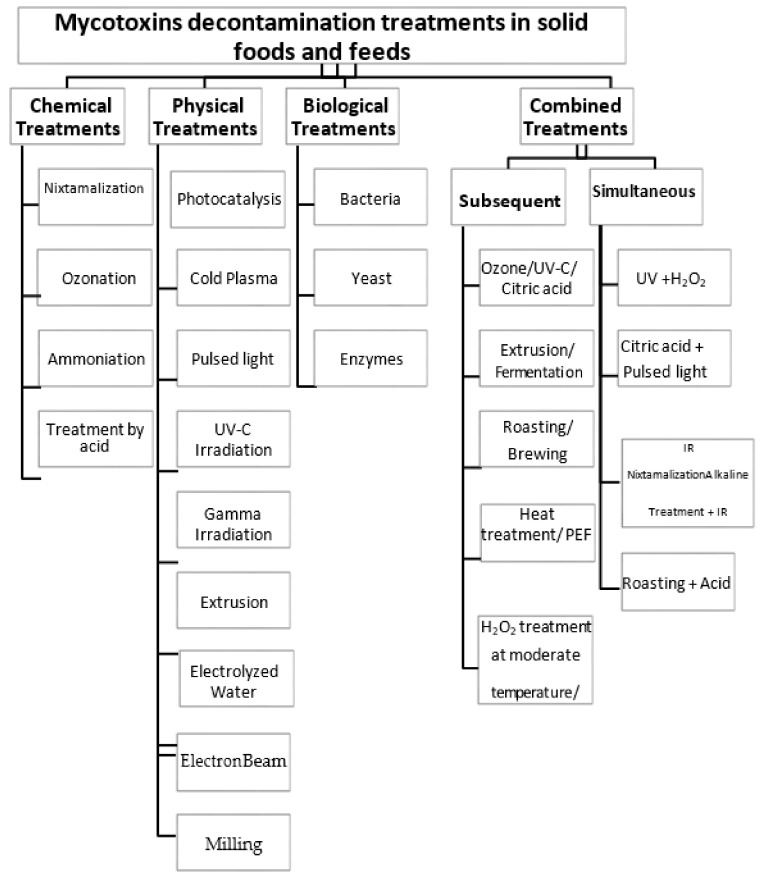
Flow chart of the different mycotoxin decontamination methods discussed in this review.

**Table 1 foods-11-03304-t001:** Chemical treatments for the reduction of mycotoxins in solid foods and feeds.

Technique	Feeds/Foods	Contaminants	Experimental Parameters	Reduction Rate	Advantages/Disadvantages	References
Ozonation	Powdered sun-dried herbs and spices	AFs	Ozone concentration = 3 ppm/time 210 min	Highest level of aflatoxin reduction: 93.75% for licorice 90% for peppermint	Advantages: Fumigation with Ozone: 3 ppm/time: 280 min— Sanitation and reduction of microbial load; Active against a wide range of microorganisms, viruses, Gram-negative and Gram-positive bacteria, spores, and fungi; Instability of Ozone—transformation into O_2_-O_3_ has a Gras status; The major biologically active constituent attributed to the medical properties of the chamomile flower was increased. Disadvantages: Reduction of chamomile essential oil by 57.14% and peppermint by 26.67%.	[44]
Ozonation	Parboiled Rice	Mycotoxins	Parboiled rice grains treated with ozone	Significant reduction of mycotoxins contamination, regardless of the time and period of application and the mycotoxin evaluated	Advantages: After soaking samples in ozone for 3 and 5 h: Higher head rice yield, luminosity and hardness, decreased cooking time, percentage of defective grains, and soluble protein.	[45]
Ozonation	Aqueous medium	TrichotheceneMycotoxins (TC)	Saturated aqueous ozone (≈25 ppm)	Degradation of TC mycotoxins to materials that were not detected by UV or MS	Disadvantages: Ozone is a toxic gas, so all preparations were conducted in a fume hood.	[46]
			At lower levels (≈0.25 ppm) of aqueous ozone	Intermediate products were observed		
			Ozonation was sensitive to pH.			
			pH 4 to 6	Maximum reduction rates		
			pH 9	No reaction		
Ozonation	Wheat	DON	↓ initial concentrations of DON solution treated with ↑ concentrations of ozone, and ↑ times	↑ DON degradation rates	Advantages: No significant changes in the protein content, sedimentation value, pasting properties, and water absorption; Improvement in the flour quality. Slight ↑ in dough development time and stability time; No decrease in the quality of wheat for end-users; Products produced from ozone-treated wheat flour (noodles) have a longer shelf life, lower darkening rate, and microbial growth; No harmful residues, easy to use, and no waste. Disadvantages: Ozone treatment in solution is faster than gaseous treatment of scabbed wheat.	[47]
			In Solution: Processing time = 30 s; Ozone concentration = 1 mg L^−1^	Degradation rate of DON = 54.2%		
			In scabbed wheat: Processing time = 12h; Moisture content = 17%; Ozone gas concentration = 60 mg L^−1^	Degradation rate of DON = 57.3%		
			Gaseous ozone	Effective against DON in scabbed wheat		
			↑ Ozone concentration and ↑ processing time	↑ Degradation rate of DON		
Ozonation	Grains	AFs	Ozone concentration = 47,800 ppm The average retention time = 1.8 min. Screw Conveyor System	Decreased *Aspergillus flavus* counts in a single pass through the screw conveyor: ↓ 96%; Reduction rate of aflatoxin: 20–30%	Advantages: Treatments with humidified and dry ozone: similar effects on fungi and insects; ↑ residence time: ↑ insect mortality and mold reduction. Disadvantages: The total electricity cost for running the equipment at maximum load was USD 3.98/h based on an electricity rate of USD 0.11/kWh; The reduction was not sufficient enough to be of commercial value; Electricity and equipment are needed.	[48]
Ozonation	Rice	Filamentous fungi	An application of 0.393 kg O_3_ m^−3^ rice	Different concentrations of ozone along the silo: 10^−1^, 10^−2^, and 10^−3^ (mol m^−3^) for the portions IP, CP, and SP, respectively;	Advantages: No damage to grain quality; No significant alteration of the quality of rice, starch modifications, lipid peroxidation, protein profile, and microstructure alterations.	[49]
				highest concentration of ozone in the inferior part of the silo at the ozone inlet = Strong fungi reduction		
Nixtamalization	Maize	AF and Fumonisins	Soaking in a solution of: 1% slaked lime ((Ca(OH)2) Or 1% traditional liquid ash	AF: up to 90%	Advantages: Increased Niacin content; Peak Viscosity: lime-nixtamalized maize flour < ash-nixtamalized flour < non-nixtamalized maize flour; Good consumer acceptability after sensorial evaluation of products; Cost-effective due to better pasting properties; Wood ash nixtamalization improves safety and quality. Disadvantages: Washing and drying steps are required at 60 °C for 16 h; Slight reduction in fat, sugar, protein, and dietary fiber content.	
				Fumonisins: up to 80%		[50]

Nixtamalization	Maize	AF	Traditional Nixtamalization Process-TNP	Not efficient enough to eliminate aflatoxins present in contaminated maize	Disadvantages: This process generates a large amount of wastewater; Possible reversibility in an acid medium such as the stomach.	[51]
Nixtamalization	Tortilla	AFB_1_	Alkaline pH of the maize-dough = 10.2, Resting time = 30–40 min of resting at room temperature	AFB1: 100%		[52]
Nixtamalization	Maize and Sorghum	FBs, DON, NIV, and ZEN	The use of 5 cooking ingredients—1 g of cooking ingredient/400 mL of water at 92 °C for 40 min		Advantages: Sodium hydroxide and potassium hydroxide are good alternatives to calcium hydroxide; Sodium hydroxide could be used in the industrial nixtamalization process. Disadvantages: Environmental concerns about using calcium hydroxide; The high pH of the byproducts and wastewater when using calcium hydroxide; Calcium chloride is not effective in reducing mycotoxins.	[53]
			Calcium chloride as a cooking ingredient	The least effect on mycotoxin reduction		
Ammoniation	Groundnut press cake	AFs	Ammoniation at (0.5–2.0%) to feed materials/moisture content: 12–16%, at 45–55 psi, and at 80–100 °C for 20–60 min	Reductions in the levels of aflatoxin of between 96% and 99%	Disadvantages: Insufficient information was available to conclude on the safety and efficacy of the proposed decontamination process; No evidence that the proposed process is sufficient to ensure irreversibility in acid medium (GIT).	[54]
Ammoniation	Wheat kernels	DON	Treatment with Ammonia vapor at 90 °C for 2 h	Degradation of DON >75%	Advantages: In silico evaluation estimated a decrease in toxicity and biological effects.	[37]
			With an initial level of DON up to 2000 μg/kg	Treatment efficacy is not affected		
Ammoniation	Corn	AFs	The use of aqua-ammonia	Effective and inexpensive	Advantages: Effective and inexpensive, and it can be applied on the farm at low cost by sealing the grain in plastic. Disadvantages: Corn treated with ammonia turns dark because the sugar (altrose) is caramelized and the grain temperature increases by about 10 °F at the time of treatment; Not an FDA-approved process and treated corn cannot be legally shipped out of state; Personal safety precautions must be taken as ammonia reacts with copper, and a motor in the air stream could cause an explosion; Corn treated with aqua ammonia requires drying for storage after treatment.	[55]

Ammoniation	Maize	AFs	The effect of ammonia	More destructive to aflatoxins G_1_ and G_2_ compared with aflatoxin B_1_ and B_2_		[56]
			Highest detoxification rate	Aflatoxins G_1_ (95%) Aflatoxin G_2_ (93%)		
			Lowest degradation rate	Aflatoxin B_1_ (85%) Aflatoxin B_2_ (83%)		
Acid	Selected Nuts	AFs	Moisture Levels: walnut (10 ± 3 and 16 ± 3%); pistachio (10 ± 3%); peanuts (10 ± 3%) Citric, Lactic and propionic acid at 9% Time: 15 min	Reduction rate of aflatoxins: citric acid (99%); lactic acid (99.9%); propionic acid (96.07%)	Advantages: Food-grade organic acids do not affect the nuts’ quality.	[57]
			Citric acid	Considerable reduction of the 4 aflatoxins; No formation of hazardous residues		
			Lactic acid	Significant reduction of AFB_1_ and Total Afs; Increase in AFB_2_ and AFG_2_; Lactic acid converts AFB_1_ into AFB_2_ (less toxic)		
			Propionic acid	More efficient to reduce AFB_1_		
Acid	Feeds/Foods	DON	5% solutions of lactic acid and citric acid	Reduction of the concentration of common trichothecene mycotoxins, especially DON and its derivate 15Ac-DON		[58]
			5% solutions of lactic acid and citric acid	No or only small effects on zearalenone, fumonisins, and culmorin		
			Lactic acid treatment	Decreased concentration of nivalenol		
Acid	-	AFB_1_	1 M citric acid—at Room temperature—Time: 96 h	conversion of AFB_1_ to AFB_2a_ >97%	Advantages: Organic acids have few detrimental effects; Under these conditions, > 71% of AFB_1_ was hydrated to AFB_2_a and did not show any reversion to the parent compound after being transferred to a neutral solution; Conversion of AFB_1_ to AFB_2a_ in a gastric environment can be enhanced by the addition of citric acid. Disadvantages: Discoloration of various types of meats including beef, pork, and fish along with minor alterations in odor and taste.	[59]
			0.1 and 1 M citric acid—at boiling temperature—Time: 20 min	Conversion of AFB_1_ to AFB_2_a > 98%		

**Table 2 foods-11-03304-t002:** Physical treatments for the reduction of mycotoxins in solid foods and feeds.

Technique	Feeds/Foods	Contaminants	Experimental Parameters	Reduction Rate	Advantages/Disadvantages	References
Photocatalysis	Wheat	DON	In solution: DON concentration = 10 μg/mL, time = 60 min, simulated sunlight: using NaYF_4_:Yb,Tm@TiO_2_ (6 mg/mL), pH = 8.0	Rate of DON degradation ≈ 100%	Disadvantages: Decreased efficiency caused by shielding effect.	[69]
			3 photocatalytic degradation products were identified	C_15_H_20_O_8_, C_15_H_20_O_7,_ and C_15_H_20_O_5_		
			In wheat:1 mL of 50 μg/mL DON standard solution + 5 g wheat-soaked and naturally dried.	Degradation rate at 120 min = 69.8%		
			Toxic grains + UCNPs aqueous solution/ratio 1:1			
			After 1 h of adsorption equilibrium, the wheat samples were illuminated by Xe lamp (200–2500 nm) for 5, 15, 30, 60, 90, and 120 min, respectively			
Photocatalysis	Wheat	DON	In wheat: The dosage of photocatalyst UCNP@TiO_2_ was 8 mg mL^−1^ Time: 90 minRatio of wheat to liquid: 1:2	Degradation rate at 90 min = 72.8%	Advantages: Little effect on the starch content, protein content, amino acid content, and fatty acid value of wheat. Disadvantages: The gluten content and pasting properties of wheat flour decreased significantly; The whiteness of wheat flour decreased, and the yellowness increased; The surfaces of starch granules were damaged to varying degrees with the prolongation of illumination time; The fatty acid value and wet gluten content and pasting properties of wheat decreased significantly during photocatalysis; The composite is easily removed by washing = low exposure dose.	[76]
Plasma	Corn	AFB_1_	CAP is generated by a Surface Barrier Discharge (SBD) system operating in ambient air, yielding RONS by a generation of non-equilibrium atmospheric pressure plasma in ambient air	Reduction rate of AFB_1_ after 60 s: 96%	Advantages: It requires less time than UV irradiation; Surface Barrier Discharge (SBD) plasma system was employed due to its practicability and scalability in the agri-food field.	[77]
			Initial concentration of AFB_1_ = 35 μg/ml	100% AFB_1_ decontamination in less than 120 s of treatment		
Plasma	Oat Flour	T-2 and HT-2	Low-pressure dielectric barrier discharge (DBD) plasma/different gases/time: 10–30 min		Disadvantages: Time-dependent effect of T-2 toxin treatment for the 4 gases; This treatment has similar thermal processing on mycotoxins such as cooking, roasting, and extrusion; Possible explanation-conversion of T-2 into HT-2 toxin and complex degradation pattern.	[78]
			Exposure to nitrogen for 30 min	The maximal reduction of T-2 toxin degradation (43.25%)		
			Exposure to nitrogen for 30 min	The maximal reduction of HT-2 toxin degradation (29.23%)		
			Mean degradation rate of T-2 toxins in all experiments	25.01%		
			Mean degradation rate of HT-2 toxins in all experiments	20.98%		
			Oxygen and air as working gas	No significant reduction of T-2 and HT-2		
Plasma	Maize	AFB_1_ and FB_1_	Pulsed dielectric barrier discharge (DBD) jet:		Advantages: Minimal impact on the organoleptic characteristics (e.g., firmness, color, pH); Minimal impact on the nutritional value of the treated foods (ascorbic acid, flavonoids); The low penetration depth of CAPP treatment is thought to limit degradation to a thin surface layer and so protects the majority of nutrients. Disadvantages: Its influence was shown to be dependent on the produce exposed, with losses in antioxidants or lipids reported; An enclosed environment is necessary to improve detoxification rates.	[79]
			Spiked maize grains are placed at 12 mm beneath plasma jet—Time = 10 min			
			Concentration of AFB_1_ = 1.25 ng/g	Degradation rate after 10 min of plasma exposure = 65%		
			Concentration of FB_1_= 259 ng/g	Degradation rate after 10 min of plasma exposure = 64%		
Plasma	Roasted coffee	OTA	Treatment with cold plasma: Imput power = 30 W/output voltage = 850 V/Helium flow = 1.5 L/min for 30 min	OTA reduction rate = 50%		[80]
			Using the brine shrimp (Artemia salina) lethality assay	Untreated roasted coffee = Toxic Treated roasted coffee = Slightly Toxic		
Pulsed Light	Red pepper powder	AFB_1_, Total AF, OTA	The highest fluence applied (9.1 J/cm^2^, 61 pulses, 20 s)	2.7, 3.1, and 4.1 log CFU/g reduction of yeasts, molds, and total plate counts (TPC), where initial microbial loads were 4.6, 5.5, and 6.5 log CFU/g, respectively	Advantages: The significant and apparent increase in total phenols. Disadvantages: Total color difference = slight difference; Proportional increase in temperature of the samples, max 59.8 °C.	[81]
			The highest fluence applied (9.1 J/cm^2^, 61 pulses, 20 s)	A maximum reduction of 67.2, 50.9, and 36.9% of (AFB_1_), (AF), and (OTA) was detected, respectively		
Pulsed Light	Solid medium	AFB_1_ and AFB_2_	PL at different initial concentrations of AFB_1_ (229.9, 30.7 and 17.8 μg/kg) and AFB_2_ (248.2, 32.2 and 19.5 μg/kg) and irradiation intensities (2.86, 1.60 and 0.93 W/cm^2^) of PL	The degradation of AFB1 and AFB_2_ followed the second-order reaction kinetic model well (R2 > 0.97); The degradation rate was proportional to the intensities of PL irradiation and the initial concentrations of aflatoxins		[82]
Pulsed Light	Rice	AFB_1_ and AFB_2_	PL treatment of 0.52 J/cm^2^/pulse for 80 s to rough rice	AFB_1_ reduction rate = 75% AFB_2_ reduction rate = 39.2%	Advantages: Safer working environment for those involved in post-harvest handling and milling operations.	[83]
			PL treatment of 0.52 J/cm^2^/pulse for 15 s to rice bran	AFB_1_ reduction rate = 90.3%AFB_2_ reduction rate = 86.7%		
UV-C Irradiation	Brown, black, and red rice (Moisture content = 13%)	Aflatoxin (B_1_,B_2_, G1, and G2), DON, OTA, and ZEN	In black and red rice–the UV-C irradiation treatment (dosage of 2.06 kJ/cm^2^) for 1 h	Effective in fungal decontamination, photo-degradation of mycotoxins	Advantages: (dosage of 2.06 kJ/cm^2^) for 1 h: ◦ Release of bound phenolics in black and red rice grains; ◦ No changes in cooking and color properties. Disadvantages: (dosage of 6.18 kJ/cm^2^) for 3 h: ◦ Reduced the total content of phenolic compounds in black and red rice; ◦ Browning grains.	[73]
			In black and red rice—the UV-C irradiation treatment (dosage of 6.18 kJ/cm^2^) for 3 h	Increased the efficiency of fungal decontamination and reduced mycotoxins		
			In brown rice, the treatment conditions need to be optimized since only the dosage of 6.18 kJ/cm^2^	Reduction of fungal contamination		
UV-C Irradiation	Maize and peanut	AFB_1_	After ten days of incubation and irradiation treatment delivering a dose of 8370 mJ/cm2	The highest reduction of *A. flavus* count was 4.4 log CFU/g in maize and 3.1 log CFU/g in peanut	Advantages: Only minimal changes in the evaluated sensory and physical characteristics (color and texture).	[84]
			Depending on the treatment	AFB_1_ reduction level:In maize ranged from 17 to 43% In peanut ranged from 14 to 51%		
UV-C Irradiation	Peanut	AFB_1_	The darkening of the UV indicator (AgCl)	Linearly proportional to the UV dosage from 0 to 120 mJ/cm^2^ delivered on peanuts	Advantages: Reducing time-consuming tasks, such as replacing manual color measurement with automatic imaging processing technology, should also be considered. Disadvantages: For scaling up the process, more parameters, such as the dimensions of peanuts, or the friction between peanuts and the chamber, should be considered; The increase was not significant and the skins remaining on hollows of peanuts could protect AFB1 from being fully exposed to UV; The AFB1 spiking process could also contribute to the result.	[85]
			Rotation at 11 rpm in the cylindrical chamber	Significant improvement in UV uniformity		
			UV irradiation: 2.3 mW/cm^2^ UV-C for 2 h with rotation at 11 rpm	Reduction percentage by 23.4% (from 14.3 ± 3.4% to 17.7 ± 4.5%)		
			UV irradiation: 2.3 mW/cm^2^ UV-C for 2 h with rotation at 11 rpm	Increased AFB_1_ degradation rate from 60.8 ± 15.3 pmol g^−1^h^−1^ to 75.0 ± 10.9 pmol g^−1^h^−1^		
Gamma Irradiation	Maize	AF and OTA	Gamma irradiation dose of 6.0 kGy	Completely inhibited the growth of the two molds		[86]
			Gamma irradiation dose of 4.5 kGy	Reduced the production of their mycotoxins		
			Gamma irradiation dose of 20 kGy	Maximum reduction rate is as follows: AFB1: 40.1% AFB2: 33.3% OTA: 61.1%		
Gamma Irradiation	Wheat flourgrape juiceandwine	OTA	In wheat flour, a radiation dose of 30.5 kGy	OTA reduction rate = 24%	Advantages: OTA was easily degraded by gamma irradiation when dissolved in water. Disadvantages: OTA is very sensitive to irradiation in water solutions but resistant in its dry form and in food matrices; Dry OTA was extremely resistant to gamma radiation.	[87]
			In grape juice, a radiation dose of 30.5 kGy	OTA reduction rate = 12%		
			In wine, a radiation dose of 30.5 kGy	OTA reduction rate = 23%		
Gamma Irradiation	Sorghum	OTA and AFB_1_	Gamma irradiation dose of 3 kGy	Sufficient to eliminate 90% of the natural fungal load of sorghum		[88]
			At a radiation dose of 10 kGy	The maximum reduction rate of AFB_1_ = 59%		
			At a radiation dose of 10 kGy	The maximum reduction rate of OTA = 32%		
Extrusion	Whole grain triticale flour	DON, 3- and 15-AcDON, HT-2, TEN, AME	Optimal parameters of co-rotating twin-screw extruder for lowering the concentration of each investigated mycotoxins in naturally contaminated flour were: SS = 650 rpm, FR = 30 kg/h, MC = 20 g/100 g	Reduction rate of mycotoxins: DON: 9.5%; 3-AcDON: 27.8%; 15-AcDON: 28.4%; HT-2: 60.5%; TEN: 12.3%; AME: 85.7%		[89]
Extrusion	Cornmeal	AF: B_1_, B_2_, G_1_, G_2_	Extrusion in the absence of high-amylose cornstarch	A reduction in aflatoxins level: (B_1_: 83.7%, B_2_: 80.5%, G_1_: 74.7%, and G_2_: 87.1%)	Disadvantages: The bioaccessibility indicates that: ◦ Part of aflatoxins reduction observed after the extrusion may be caused by their interactions with food matrix macromolecules; ◦ Once the digestion is completed, part of these toxins becomes available for absorption in the small intestine.	[90]
			Extrusion in the presence of high-amylose cornstarch	Higher aflatoxins reductions were observed: (B_1_-89.9%, B_2_-88.6%, G_1_-75.0%, and G_2_-89.9%)		
Electrolyzed Water	Wheat grains	DON	For AcidEW		Advantages: Both pH 5.5 AcidEW and pH 9.5 AlkEW did not change the basic properties of wheat, including whiteness, moisture content, crude protein content, and wet gluten content; No remarkable change in isolated starch morphology; AcidEW can improve the farinograph property of wheat flours with higher stability time and FQN and a lower degree of softening; AcidEW is a promising way for large-scale wheat milling operations to eliminate DON and mycological contaminations in wheat grains.	[91]
			pH 5.5	Optimal pH for DON elimination		
			pH 2.5	Optimal pH for fungal reduction		
			For AlkEW			
			pH 9.5	Optimal pH for DON elimination		
			pH from 8.5 to 12.5	Strong elimination activity on fungi		
Electron Beam	Red pepper powder	OTA	Treatment at 6 kGy	Reduction of yeast count by 3 log CFU/g Reduction of mold count by 4.4 log CFU/g	Advantages: Retention of more than 85% of total phenols, carotenoids, and antioxidants activity; No detrimental effect on the physicochemical quantities of the red pepper powder. Disadvantages: Slight color differences.	[92]
			Treatment at 10 kGy for 23 s	Reduction of total plate counts by 4.5 log CFU/g		
			Treatment at 30 kGy	Reduction rate of OTA: 25%		
Milling	Maize	Mycotoxins	Grain cleaning	Reduction of fungal metabolites by 1.2–2 times	Advantages: Flaking grits is the healthier milling product of maize with the lowest mycotoxins content. Disadvantages: Highest mycotoxins content in animal feed products threatening animal health; Redistribution of mycotoxins in maize fractions after milling; Most mycotoxins are concentrated in the germ.	[93]
Milling	Maize	B-series fumonisins (FBs)	Grain cleaning	Reduction rates of FBs: 42%	Advantages: Higher reduction rates of FBs are achieved by tempering degermination; The separation of horny endosperm from the fine fractions is better in the tempering process; Higher reduction rates are achieved in the highest-sized flaking grits. Disadvantages: The germ and animal feed flours contain a higher amount of FBs than other milling products.	[94]
			Dry-degermination process of uncleaned kernels	Reduction rates: Maize flour: 50% Break meal: 83% Pearl meal: 87%		
			Tempering degermination Process of uncleaned kernels	Reduction rates: Small grits: 78% Medium grits: 88% Flaking grits: 94%		

**Table 4 foods-11-03304-t004:** Subsequent techniques to mitigate mycotoxins in solid foods and feeds.

Combination	Feeds /Foods	Contaminants	Experimental Parameters	Reduction Rate			Advantages/Disadvantages	References
Ozone/UV-C/Citric acid	Pistachio nuts	Aflatoxins	Combination of the immersion of the samples in 3 N CA, 30 min exposure to O3, and 36 h exposed to UV-C radiation	AFB_1_ and AFB_2_ > 90% AFG_1_ and AFG_2_ > 99%			Advantages: No significant changes were observed in total fat content, protein content, acid and peroxide content, total phenolic compounds, soluble and insoluble carbohydrates of pistachios; No significant changes between sweetness, acidity, flavor, color, and overall quality of treated and non-treated samples; The combination of O_3_, UV-C, and CA was much more effective than the effect of each of them alone.	[123]
			The UV-C	More effect on AFB_2_ and AFG_2_				
			The O_3_ treatment	Degradation of AFB_1_ and AFG_1,_ more than AFB_2_ and AFG_2_				
			Acid treatment	More effect on AFB_1_ and AFG1, against AFB_2_ and AFG_2_				
Extrusion/Fermentation	Wheat bran	Mycotoxins	Extrusion at 130 °C—Screw speed: 20 rpm + fermentation with *L. casei* and *L paracasei* strains at 30 ± 2 °C for 24 h.	The lowest overall concentration of the tested mycotoxins.			Advantages: Appropriate extrusion parameters and LAB strain selection lead to the higher formation of L-(+)-lactic acid and lower WPBP microbial contamination (except for the M/Y count); Extrusion, as well as extrusion in combination with fermentation, reduces the total biogenic amines content (by 2 times on average).	[124]
			Extrusion at 130 °C—Screw speed: 25 rpm + fermentation with *L. casei* and *L paracasei* strains at 30 ± 2 °C for 24 h.					
Roasting/Brewing	Coffee	AF and OTA	Treatment	OTA reduction			Advantages: Roasting is performed at a suitable low temperature to preserve the traditional color and flavor.	[125]
				Low roasting:	Medium roasting:	High roasting:		
			After roasting	15.17%	46.78%	57.43%		
			After reduction after brewing	43.57%	4.11%	7.28%		
			After roasting and brewing	58.74%	60.88%	64.7%		
			Treatment	AF reduction				
				Low roasting:	Medium roasting:	High roasting:		
			After roasting	31.98%	46.36%	61.52%		
			After brewing	10.19%	1.50%	0.86%		
			After roasting and brewing	40.18%	47.86%	62.38%		
Heat treatment/PEF	Artificially spiked potato dextrose agar (PDA)	AFT—AFB_1_	Thermal process:				Disadvantages: The rate of degradation of aflatoxin increases with an increase in the moisture content of heated food, the potato dextrose agar used as a model system for the study was of high moisture content; When the optimized parameters were adapted to the real food matrix, the degradation percentage of the toxin may vary with its moisture content.	
			At pH 10	The effect of process time was observed to affect both AFB1 and AFT content more significantly than the temperature.				[126]
			At pH 4 and 7	The effect of temperature on toxin reduction was more evident.				
			PEF: Fixed parameters: pulse frequency (50 Hz), burst (10), energy (1 KJ) Time: 10 s					
			Variables: pulse width (ms) and output voltage (%), and pH of the PDA/different combination. (20 μs 10%; 51 μs 26% and 65 μs 26% for pH 4, 7, and 10 respectively)	Reduction rates of AFB_1_: 79–96%				
			Combined effect of Thermal process + PEF:					
			Thermal process: T = 110 °C +t = 15 min + PEF (65 μs 26%, pH 10)	The maximum degradation: AFB1 = 96.881%; AFT = 95.721%				
High Conc. H_2_O_2_ at Moderate temperature/roasting	Peanuts	AF	30 g/hg H_2_O_2_ at 20 °C	AF reduction rate = 30%			The oil quality was not seriously affected by the treatment; The weight loss and oil quality change of the treated peanuts were negligible; Peanuts were able to keep intact after the treatment because the temperature of the treatment was lower than that of the starch gelatinization; Eco-friendly process; Most H_2_O_2_ was removed by drying H_2_O_2_-treated peanuts at 35 °C for 12 h.	[127]
			30 g/hg H_2_O_2_ at 50 °C	AF reduction rate = 73%				
			30 g/hg H_2_O_2_ at 50 °C for 8 h —unroasted peanuts	AF reduction rate = 86%				
			Combined effect: 30 g/hg H_2_O_2_ at 50 °C for 8 h + roasted peanuts at 140 °C for 10 min	AF reduction rate = 90%				

**Table 5 foods-11-03304-t005:** Simultaneous techniques to mitigate mycotoxins in solid foods and feeds.

Combination	Feeds/Foods	Mycotoxins	Experimental Parameters	Reduction Rate	Advantages/Disadvantages	References
UV + H_2_O_2_	Peanuts	AFs	Advanced Oxidation Processes by UV and H_2_O_2_		Advantages: The AOP treatment can be considered environmentally friendly; No waste—the degradation compounds are only water and oxygen; This combination accelerates the degradation rate of AF; Drying peanuts at 35 °C for 12h: complete removal of residual H_2_O_2_. Disadvantages: The oil quality was slightly affected by the AOP treatment in whole kernels, but a more severe influence on oil quality was observed in the milled kernels; The color of whole kernels slightly darkened but not considerably affect its appearance.	[128]
			1 h AOP (2.76 mW/cm^2^ UV-C, 1 g/hg H_2_O_2_) of whole peanut kernels	Degradation rate of AF = 33%		
			1 h AOP (2.76 mW/cm^2^ UV-C, 1 g/hg H_2_O_2_) of milled kernels	Degradation rate of AF = 60%		
Citric acid + Pulsed light	Peanuts	AFB	PL + CA treatment	AFT ≈ 98.2% AFB_1_ ≈ 98.9% AFB_2_ ≈ 98.1%	Advantages: No significant changes in chemical quality. Disadvantages: Significant changes in color.	[129]
IR Nixtamalization (alkaline treatment + IR)	Maize tortillas	AFs	The infrared nixtamalization process (IRNP)—Cooking in a cooker that generates infrared radiation (14.2 A, 1704 W)	The degradation rate of AF: 93.82%	Advantages: They did not show adduct AFB1-Lys formation; An effective method for aflatoxin detoxification in maize tortillas, as it generates degradation products less toxic than those used in traditional nixtamalization.	[130]
			Traditional nixtamalization process (TNP)	The degradation rate of AF: 98.35%		
Roasting + acid	Pistachio nuts	AFB_1_	Treatment 1: 50 g Pistachio—addition of 30 mL water + 30 mL lemon juice + 6 g of citric acid—roasting at 120 °C for 1 h	AFB_1_ = 93.1 ± 8.2%	Advantages: Useful and safe degradation method of AFB1 in naturally contaminated pistachio nuts; Treatment 2 caused no noticeable change in the desired appearance of pistachios. Disadvantages: Treatment 1 altered the desired physical properties.	[131]
			Treatment 2: 50 g Pistachio—addition of 30 mL water + 15 mL lemon juice + 2.25 g of citric acid—Roasting at 120 °C for 1 h	AFB_1_ = 49.2 ± 3.5%		

## Data Availability

Data is contained within the article.

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
