# Peer review of "Single, Subsequent, or Simultaneous Treatments to Mitigate Mycotoxins in Solid Foods and Feeds: A Critical Review"

_foods, 2022, doi:10.3390/foods11203304_

Round 1
Reviewer 1 Report
In this article, the authors review the various techniques used in the food industry to eliminate mycotoxins potentially present in solid foods. First, the authors present the chemical, physical and biological treatments applied separately. A second part presents the cumulative or synergistic effects of different combined treatments applied one after the other. Finally, the authors are interested in the results obtained during simultaneous treatments using different physico-chemical techniques. Each part includes a well-constructed table which summarizes the characteristics of each technique, the mycotoxin reduction rates obtained and for each treatment presents the advantages and disadvantages. These tables are particularly welcome because the article remains quite descriptive and tends to be a catalog of techniques, which sometimes makes reading it a little tedious.
Overall, this is important (116 publication references) and relevant work for compiling data on treatments aimed at reducing the mycotoxin content of food. The major criticism, inherent in this kind of review, lies in the catalog aspect mentioned above.
Minor points:
L171: change “concurrently” to “Concurrently”
L184: change “that in which” to “in which”
L188/189: The sentence “Ammonia is more effective against…” is not a hypothesis but an affirmation.
L196/198: This sentence is unclear and should be rewritten
L221: botthacids? It is probably “both acids”
Author Response
Dear Reviewer
Please see the attachement

Reviewer 2 Report
This paper is very well written and it is very useful to have more papers summarizing mitigation measures for frequently detected and toxic compounds such as mycotoxins.
To improve the paper the following general comments should be changed/corrected:
Figure 1.- it may be better to divide physical treatments into invasive and non-invasive. Some treatments were not included in Figure 1. as well as in the Manuscript: Physical (milling and cleaning- which are indispensable techniques in the processing of solid food and feed, and there are a lot of papers about their implication for mycotoxins reduction; the application of high and low temperatures is not discussed); Non-invasive treatment (electron beam)
“The most commonly known mycotoxins to contaminate foods and feeds are the following: aflatoxins AFs (AFB1, AFB2, AFG1, AFG2, AFM1), ochratoxin (OTA), trichothecenes (deoxynivalenol: DON, nivalenol: NIV, T-2 toxin: T-2, HT 2 toxin: HT-2), zearalenone (ZEN), fumonisins B1 (FB1), enniatins (EN) “- Since authors listed the both, regulated and non-regulated mycotoxins, in this sentence there are not included some mycotoxins that are recognized as the most frequently detected contaminants of solid food and feed (other fumonisins, moniliformin, and other emerging mycotoxins).
According to the Food and Agriculture Organization (FAO) reports, 25% of the crops in the world are contaminated by mycotoxins [13].- In the reference list, reference at number 13. Is Mahato et al., On this place, it should be FAO reference. However, data that 25% of the crops in the world are contaminated by mycotoxins, is overcome. Please find some newer references for this claim. For example: Eskola, M., Kos, G., Elliott, C. T., Hajšlová, J., Mayar, S., & Krska, R. (2020). Worldwide contamination of food-crops with mycotoxins: Validity of the widely cited ‘FAO estimate’of 25%. Critical reviews in food science and nutrition, 60(16), 2773-2789.
“Total aflatoxins limits in food has 58 been established in 2003 in 48 countries to be in the range of 0 to 5 mg/Kg”- Are You sure that units are written properly, to my opinion mg/Kg should be changed with µg/kg. 5 mg/Kg is a very high concentration for the maximum level in food.
Conclusion/Discussion: Please add some sentences related to the great need for more results in terms of the investigation of the application of mitigation techniques on samples that are naturally contaminated with a great number of different mycotoxins. Please also add your opinion about future needs, because a lot of papers published in recent years reported that some food and feed materials contained several dozen different mycotoxins (especially cereals). Further, the lack of a great number of published papers investigating mitigation measures is the application of technique 1. on spiked samples instead of naturally contaminated; 2. In laboratory conditions on small-size samples, which are further difficult to apply and test their effectiveness in real industrial-type systems.
Reviewer 3 Report
This manuscript reviewed some chemical, physical and biological treatments to reduce some typical mycotoxins, such as AFs, DON, t-2, in solid grains and feeds. Since the ubiquitous prevalence of mycotoxins in food and feed caused grains and cereals waste, and human and animal health problems, a large number of studies on eliminating mycotoxins and their hazards have been carried out. This article focuses on the elimination of aflatoxin, and the title is mycotoxin, which is obviously not appropriate. Secondly, This manuscript lists the results of in vitro tests to evaluate the elimination effect of mycotoxins, which is one-sided. Because the metabolites of mycotoxins can be reversibly transformed into toxin prototypes after treatment or in vivo, the author should clarify the degradation products of mycotoxins and the mechanism of elimination of the mycotoxins of each treatment, and supplement the toxicity or safety of these treated foods or feedstuffs compared with untreated foods or feedstuffs in vivo by experimental animals. The article lacks the literature and elaboration of this important aspects.Thirdly, authors should better to evaluate the cost of these treatment methods and the possibility of their application in grain and feed production.
Reviewer 4 Report
The manuscript titled “Single, subsequent, or simultaneous treatments to mitigate mycotoxins in solid foods and feeds: A critical revie” was reviewed for consideration in Foods. The topic is interesting and according to the scope of the Journal. However, following points should be considered before acceptance.
Line 33: Please correct the numbering style of references. And throughout the manuscript correct the style
Line 497: Which one is the Table 16.4?? please correct
All five tables 1-5, the country should mention in separate column, because mycotoxins happened to be more dangerous in tropical and subtropical regions.
Conclusion
The conclusion part should be revised and main highlights concluded from manuscript should only be stated.
